# A First Methodological Development and Validation of ReTap: An Open-Source UPDRS Finger Tapping Assessment Tool Based on Accelerometer-Data

**DOI:** 10.3390/s23115238

**Published:** 2023-05-31

**Authors:** Jeroen G. V. Habets, Rachel K. Spooner, Varvara Mathiopoulou, Lucia K. Feldmann, Johannes L. Busch, Jan Roediger, Bahne H. Bahners, Alfons Schnitzler, Esther Florin, Andrea A. Kühn

**Affiliations:** 1Movement Disorder and Neuromodulation Unit, Department of Neurology, Charité Universitaetsmedizin Berlin, 10117 Berlin, Germany; 2Institute of Clinical Neuroscience and Medical Psychology, Medical Faculty, Heinrich Heine University Düsseldorf, 40225 Düsseldorf, Germany; 3Department of Neurology, Center for Movement Disorders and Neuromodulation, Medical Faculty, Heinrich-Heine University Düsseldorf, 40225 Düsseldorf, Germany

**Keywords:** Parkinson’s disease, bradykinesia, finger tapping, accelerometer, open-source, machine learning, motor monitoring, symptom prediction

## Abstract

Bradykinesia is a cardinal hallmark of Parkinson’s disease (PD). Improvement in bradykinesia is an important signature of effective treatment. Finger tapping is commonly used to index bradykinesia, albeit these approaches largely rely on subjective clinical evaluations. Moreover, recently developed automated bradykinesia scoring tools are proprietary and are not suitable for capturing intraday symptom fluctuation. We assessed finger tapping (i.e., Unified Parkinson’s Disease Rating Scale (UPDRS) item 3.4) in 37 people with Parkinson’s disease (PwP) during routine treatment follow ups and analyzed their 350 sessions of 10-s tapping using index finger accelerometry. Herein, we developed and validated ReTap, an open-source tool for the automated prediction of finger tapping scores. ReTap successfully detected tapping blocks in over 94% of cases and extracted clinically relevant kinematic features per tap. Importantly, based on the kinematic features, ReTap predicted expert-rated UPDRS scores significantly better than chance in a hold out validation sample (n = 102). Moreover, ReTap-predicted UPDRS scores correlated positively with expert ratings in over 70% of the individual subjects in the holdout dataset. ReTap has the potential to provide accessible and reliable finger tapping scores, either in the clinic or at home, and may contribute to open-source and detailed analyses of bradykinesia.

## 1. Introduction

Parkinson’s disease (PD) is a neurodegenerative movement disorder, afflicting nearly 10 million people worldwide, with the number of diagnoses expected to increase substantially in the coming years (e.g., 1.6 factor increase by 2050) [1,2,3,4]. Bradykinesia and akinesia, defined as slowness and lack of movement initiation, respectively, are cardinal symptoms of PD and negatively impact the quality of life in people with PD (PwP) [5]. For example, one such impairment contributing to poor quality of life in PwP is a bradykinesia-induced decrement in fine motor control of the hands, causing individuals to lose their ability to perform rudimentary daily activities such as handwriting, brushing teeth, or even buttoning a shirt [6]. Typically, treatment for bradykinesia in PD consists of pharmacological therapies to restore the pathologically depleted extracellular dopamine levels in the striatum [7]. However, long-term dopamine replacement therapy, parallel to disease progression, eventually leads to motor fluctuations (e.g., diminishing therapeutic effects, shorter periods in optimal medication conditions, and more frequent periods with severe bradykinesia or dyskinesia in between medication intakes) in approximately 50% of all PD patients [8]. Such motor fluctuations have a large burden on patients and caregivers alike, and, importantly, are often a clinical indication for advanced therapies that may also be required for more optimal motor symptom reduction (e.g., duodenal levodopa infusion or deep brain stimulation (DBS)) [9,10,11]. Thus, motor fluctuation assessment is an essential part of PD clinical care, and valid automated, technology-based solutions for characterizing clinical features of bradykinesia would substantially improve the reliability, reproducibility, and accessibility of motor symptom assessments in PD [12,13,14]. Despite technological and computational advances in movement monitoring, bradykinesia assessments, in practice, still largely depend on labor-intensive and subjective physical examinations by expert raters (e.g., Unified Parkinson’s Disease Rating Scale: UPDRS Part III Motor Examination), which often yield poor reliability and reproducibility [15,16,17,18]. Thus, there remains a need for developing quantitatively derived estimates of motor fluctuations in order to complement existing gold standards for symptom monitoring in PD.

An accurate and reliable motor assessment tool for PD would ideally provide reliable symptom severity scores per category (e.g., bradykinesia, tremor, gait disorders, dyskinesia, etc.) based on passive movement monitoring (e.g., general changes in body movement frequency/speed), which would not require the individual to perform structured motor tasks [15]. However, the development of such passive, naturalistic bradykinesia monitoring in short time windows (e.g., on the order of minutes to an hour) is especially challenging compared to other symptom subtypes in PwP (e.g., tremor) [19,20,21]. In contrast, an open-source, validated, and easy-to-use bradykinesia assessment tool would allow clinicians to profit from task-relevant motor fluctuation monitoring without reliance on subjective clinical ratings. Moreover, the emerging possibilities of collecting other chronically monitored physiological data (e.g., subcortical local field potentials, heart rate, or sleep metrics) via sensing-enabled devices (e.g., DBS pulse generators or smartphones and -watches) further underscores the timely relevance of simultaneous behavioral monitoring in order to aid symptom and therapy-related assessments in PwP [22,23]. Importantly, task-relevant assessments of bradykinesia which are feasible to perform multiple times per day have the potential to support the further development of passive movement monitoring approaches. They can provide information on task-specific symptom severity, and also potentially reduce or replace lengthy in-person clinical visits and/or labor-intensive training periods that are currently required for passive monitoring [20,21,24,25,26].

Proposed methods for objective, task-related bradykinesia assessments often make use of accelerometry, video-based motion capture, and keyboard- or smartphone-based tapping tasks (for a review comparing movement monitoring devices for bradykinesia monitoring) [27,28,29,30,31,32,33]. Overall, video-based recordings of movements are useful in predicting expert-rated UPDRS bradykinesia symptoms (both single and composite bradykinesia scores) with intraclass correlation coefficients (ICC) around 0.7–0.8 [30,32]. However, these methods require excellent self-recording by the individual and/or investigator, and the algorithms are often proprietary or not validated in an external or holdout validation sample, limiting comparability and reproducibility. Similarly, keyboard- and smartphone-based methods using finger tapping tasks report overall correlations with UPDRS sub scores around 0.4–0.5, with excellent performance exceptions of rho circa 0.8 also observed [28,29,31]. In contrast, despite the accessibility and the relatively low cost of accelerometry, there are no validated, automated, open-source accelerometer-based algorithms published so far to the best of our knowledge. 

Therefore, we aim to fill this gap by developing and validating an open-source algorithm, ReTap, which provides automated bradykinesia assessment using a UPDRS-based finger-tapping task (i.e., tapping scores, according to UPDRS Part III Item 3.4). Of note, finger-tapping assessments were chosen as our primary focus for this algorithm, as finger-tapping performance may reflect reliable markers of general bradykinesia-related impairments in motor function for movement disorder patients (e.g., PwP, progressive supranuclear palsy, dystonia, ataxia) [33,34]. ReTap’s algorithm first detects blocks of tapping activity, as well as single-trial taps, in raw or pre-processed accelerometer (acc) data. Next, it extracts clinically relevant kinematic features (e.g., indices of movement amplitude, frequency, variability, and their decrement) to predict expert-rated UPDRS Part III Item 3.4 finger-tapping scores using a random forest classification that was validated in an unseen holdout dataset. By providing validated UPDRS Part III Item 3.4 score predictions, as well as relevant kinematic features for movement blocks automatically, ReTap has the potential to support accessible, out-of-hospital motor fluctuation monitoring (i.e., tracking of treatment responses and symptom progression) for PwP in the future.

## 2. Materials and Methods

### 2.1. Study Sample

We studied PwP who were originally enrolled as part of larger projects examining motor network dysfunction (“Retuning dynamic motor network disorders using neuromodulation,” TRR295-424778381) from two academic movement disorders clinics in Düsseldorf and Berlin, Germany (for relevant demographic information, see Appendix A). Our inclusion criteria required patients to have a PD diagnosis and that they were treated with both dopamine replacement medication and DBS at the time of study enrollment. Subjects who were not able to perform finger tapping due to comorbidities were excluded. Moreover, individuals with a history of other neurological or psychiatric disorders, severe cognitive impairment, or depression were excluded from the study. All participants gave informed consent to the locally approved study protocols (Düsseldorf: No. 2019-626_2 approved by the medical ethical committee of the University Hospital Düsseldorf, Berlin: Protocol EA2/256/20 approved by the medical ethical committee of Charité Universitaetsmedizin Berlin).

### 2.2. Accelerometer Data Recording Protocol

PwP performed finger tapping tasks in clinically defined therapeutic conditions (i.e., ON and OFF clinically effective medication (med) and stimulation (stim)). Specifically, PwP performed a unilateral finger tapping task for ten seconds. Start and stop times were verbally indicated by the instructor. Participants were seated in a chair and instructed to “raise their hand and to perform index-to-thumb taps as largely and quickly as possible”, according to the UPDRS Part III Item 3.4 instructions. Participants recorded in Düsseldorf performed one unilateral tapping sequence with their right hand per therapeutic condition. Participants recorded in Berlin performed three unilateral tapping sequences with their left and right hand per therapeutic condition. Where applicable, each tapping sequence was preceded by at least 10 s of rest.

Data were collected with a tri-axial accelerometer mounted on the distal part of the index finger. The accelerometer collected data through a digital amplifier with sampling frequencies ranging from 250 to 5000 Hertz (Hz) (Berlin: TMSi Saga or TMSi Porti, TMSi International, Oldenzaal, NL; Düsseldorf: ADXL335 iMEMS Accelerometer, Analog Devices Inc., Norwood, MA, USA recorded using Elekta/MEGIN System, MEGIN, Helsinki, Finland). All tapping tasks were simultaneously recorded with a standard video camera.

### 2.3. Clinical Motor Symptom Assessment

Clinical ratings of motor symptom severity were provided for each tapping sequence by one experienced rater (resp. RS, VM, and JH) according to the UPDRS Part III Item 3.4 recommendations.

### 2.4. ReTap Algorithm

The ReTap algorithm consists of five major parts: (i) raw accelerometer data preprocessing, (ii) active tapping block detection, (iii) single tap event detection within a tapping block, (iv) kinematic feature extraction per tapping block, and (v) prediction of UPDRS Part III Item 3.4 tapping score based on the extracted kinematic features. Importantly, although ReTap does not require any prior preprocessing of raw accelerometer signals, it is optimized to process both raw and preprocessed tri-axial accelerometer traces that do not contain other movement tasks. We will describe all functionality of the algorithm in detail (see Section 2.4.1, Section 2.4.2, Section 2.4.3, Section 2.4.4 and Section 2.4.5 below) and refer to the publicly available code on github.com/jgvhabets/ReTap (accessed on 27 April 2023) for all syntax-related details of the algorithm [35].

#### 2.4.1. Raw Accelerometer Data Preprocessing

First, ReTap resamples the raw tri-axial time series, if necessary, to 250 Hz to create uniform data samples across recording sites and to facilitate translation to future studies in this area. A sampling frequency of 250 Hz was chosen to maintain sufficient samples per tapping event, assuming that PwP usually tap 1–5 times per second. A bandpass filter between 2 and 48 Hz was then applied to detrend the data and remove 50 Hz line noise. To rectify potential differences in the order of magnitude between traces due to variations in recording equipment, the preprocessing function controls for an order of magnitude in g (i.e., m/s^2^). The model detects the orientation of typical double sinusoid accelerometer patterns and automatically inverts the time series in the case of flipped patterns where appropriate. In addition, the function detects potential noise- or movement-related artefacts (e.g., samples larger than 10 ∗ 99th percentile) and replaces them with missing values. 

#### 2.4.2. Active Tapping Block Detection

During the tapping block detection, the algorithm segments every second of data in the accelerometer trace into eight non-overlapping windows and calculates their percentage of activity (i.e., activity-%). For this, we calculate the signal vector magnitude (SVM) as a fourth time series by taking the square root of the sum of the squared values of the x-, y-, and z-sample. The activity-% equals the part of a segment (i.e., 125 milliseconds (ms)) that exceeds an activity threshold (i.e., the SVM standard deviation (sd) * 0.5). These thresholds were determined empirically based on visual inspection of the final analyzed cohort. Next, a sliding, non-overlapping window of 10 segments (i.e., 1.25 s) will label a respective time window as active if more than two segments had an activity-% of more than 30%. Finally, the detection function merges active windows closer than 2 s and afterwards discards active windows shorter than 0.32 s. The function plots the block detection result per acc-trace for visual inspection.

#### 2.4.3. Single Tap Event Detection

To calculate clinically relevant kinematic features per tap, the algorithm detects all single tapping movements within the aforementioned tapping block. We define a tapping movement here as the period between two consecutive closings of the index finger and thumb. The closing of the index finger on the thumb (i.e., the moment that the index finger touches the thumb) causes a sharp positive peak in the accelerometer trace due to the relatively large deacceleration of downwards movement, described as the contact force by Okuno et al. [36]. We will refer to this moment as the ‘impact’ moment, and the model uses this characteristic acc-peak of the impact to identify the moments where the index finger touches the thumb and to define the ending of one tap and the beginning of the next tap. To find the impact moments, the algorithm first finds all peaks in the SVM-signal exceeding a threshold (i.e., 20th percentile of the maximum (max) of the SVM signal). Second, the model excludes peaks where the first differential of the SVM signal did not exceed a certain threshold (i.e., ±20th percentile of the max or the minimum (min) of the differential signal, respectively). Finally, probable tapping peaks were required to be at least 166 ms apart from one another. The function plots all detected impact moments and the acc-trace per block for visual inspection (for visualization, see Figure 1).

#### 2.4.4. Kinematic Feature Extraction per Tapping Block

To enable the machine learning prediction of finger tapping scores, ReTap extracts several kinematic features per tapping block. These features are the input vectors for the machine learning classification models. Moreover, ReTap stores the kinematic features, per single tap and per tapping block where appropriate, to enable more detailed analyses of motor symptom fluctuations outside the use of this algorithm alone. ReTap extracts the following features across the tapping block: the total number of taps, the tapping frequency (in taps per second), tap duration (in seconds), the normalized root mean square (i.e., normed to tap duration in seconds; SVM-RMS in g), and the Shannon’s entropy (in arbitrary units, a.u.). Additionally, ReTap extracts the following kinematic features per single tap (one tap is defined as the period between two consecutive index finger-to-thumb closings): inter-tap-interval (i.e., the duration between two tap-starts; ITI), normalized SVM-RMS of the full tap (in g), SVM-RMS around the impact (in g), the velocity during finger raising in m/s, the jerkiness (as the number of directional changes i.e., rate of change of acceleration in m/s^3^), and the entropy (representing the stability and predictability of the signal, in a.u.). We defined the period of finger raising as the positive acc-peak between an impact moment (start of tap raise) and the end of the first sinusoid pattern (end of upwards movement). From all single-tap features, the model calculates the following single values per tapping block: mean, coefficient of variation (coefVar), and the decrement (i.e., the linear slope in each feature as time elapsed during the tapping block). For entropy and ITI, we used the absolute decrement value.

These kinematic features were chosen based on their performance in previous studies of accelerometer-based tapping assessments [19,20,33,37,38] (for a recent review comparing kinematic tapping features, see [27]). The rationale behind the feature selection was that they represent the clinically relevant kinematic concepts of the UPDRS Part III Item 3.4 rating instruction, namely evaluating changes in tapping frequency, tapping amplitude, and the consistency and decrement of movement amplitude and pacing over the course of the task [16]. For details regarding the computational formulas of the features, we refer to our publicly available code [35]. 

#### 2.4.5. Development and Validation of Tapping Score Prediction Model

To ensure the statistical validity and reproducibility of our predicted UPDRS Part III Item 3.4 finger tapping scores, we split the included dataset into a development (75%) and validation (25%) dataset. Creating a validation dataset which is not used during the development of the algorithm enables a true validation of the model on unseen data and is good practice in predictive analysis. A data split with equal distributions of tapping score values (i.e., UPDRS Part III Item 3.4 scores) and clinical site of recording (i.e., Düsseldorf and Berlin) was found with the help of an iterative function in the development and validation datasets. Importantly, all recording sessions from a single subject were included in either the development or the validation dataset to ensure independence between the two datasets. We developed ReTap’s algorithm using the development dataset with a cross-validation that stratified the tapping scores in different folds. Finally, we trained our final model on the full development dataset, which was then validated in the holdout validation dataset. Of note, as there were too few tapping blocks expert-rated as a 4 in our total dataset (0.3%), we excluded all tapping blocks rated as a 4 from classification analysis.

As a first step in the classification model, we classify tapping blocks with less than nine detected taps as a score of 3. In practice, this step classifies tapping blocks with very few or very small amplitude taps as a score of 3, since ‘tapping-like movements’ with very small amplitudes are not always detected by the single tap detection (see right panel in Figure 1B). This was meant to adhere to current UPDRS rating recommendations, which categorize finger tapping item-scores of 3 as amplitude decrements occurring near the beginning of the tapping block or very slow movement, i.e., very few taps. We thereby assume that our single tap detection successfully detects the majority of taps, which can be confirmed based on visual inspection of the processed accelerometer traces (see Figure 1). To prevent the classification of blocks with a few, large, well-performed taps as item-scores of 3, the algorithm will return blocks with few taps, but exceeding an empirically defined finger raise velocity threshold, to the classification model for regular tapping score prediction.

The core of ReTap’s classification paradigm is the machine learning-classification based on all extracted kinematic features (see Section 2.4.4 above). The kinematic features are the basis of the classification model and contain clinically relevant information that may differentiate motor fluctuations in finger tapping performance. Therefore, by evaluating the predictive performance of such kinematic-based UPDRS Part III Item 3.4 tapping scores, it may help scientists and clinicians to assess finger tapping in a more objective, systematic fashion. Specifically, we tested several classifiers and found that Random Forest classification (RF) was superior to Logistic Regression, Support Vector Machines, and Linear Discriminant Analysis classifiers based on relevant performance metrics (see Statistical Evaluation below). Furthermore, we compared predictive performances of classification models using features derived from the first 15 detected taps in the sequence versus all possible detected taps in the task block in order to evaluate the generalizability of ReTap’s performance, regardless of instruction set (see Appendix A).

### 2.5. Statistical Evaluation

First, to ensure equivalent distributions of expert-rated tapping scores in the development and validation datasets identified herein, we conducted a non-parametric one-way analysis of variance (Kruskal-Wallis) to test equal distributions of UPDRS Part III Item 3.4 scores. Next, in order to determine the statistical validity of our classification model, we reported the model’s predictive performance in mean prediction error expressed in raw UPDRS Part III Item 3.4 scores (ranging from 0 to 3) and the Intraclass Correlation Coefficient between the predicted and true, expert-rated tapping scores (ICC, two-way mixed effect model for k-different raters, ICC-3k) [39]. Moreover, we report the Pearson correlation coefficient between expert-rated and predicted UPDRS Part III Item 3.4 scores. With the selected metrics and the reported multiclass confusion matrix, we assess predictive performance robustly and transparently while respecting the multiclass and naturally unbalanced nature of UPDRS Part III Item 3.4 tapping scores [40].

Significance testing of the mean prediction error and the ICC-3k was done with a random-labels permutation test (n = 1000), in which we randomly shuffled the true-labels (expert-rated tapping item-scores) and repeated the prediction. Preserving the tapping score distribution in the permutation test instead of using a chance level distribution (one out of four categories, 25%) improved the validity and robustness of our significance testing. Significance levels of 0.05 were applied following Bonferroni correction for multiple comparisons.

To assess ReTap’s ability to capture intra-individual symptom fluctuations, we analyzed the expert-rated and predicted tapping-scores per individual within the validation dataset separately. We extended this analysis by testing individual feature fluctuations between therapeutic conditions for significance. We compared the mean feature values in the medication-OFF and stimulation-OFF conditions both with all other conditions (medication-ON, stimulation-ON; medication-ON, stimulation-OFF; and medication-OFF, stimulation-ON), as well as with the best ON-condition (defined as the condition with the lowest mean tapping-scores). We considered the five most important features of the RF-classifier. We tested statistical significance using Mann-Whitney-U analyses and Bonferroni-corrected *p*-values for multiple comparisons.

Finally, we reported the relative importance of each kinematic feature within the RF classifier based on the Mean Decrease Impurity method [41]. Briefly, this method represents how often each feature is used within the classification model to split between different nodes and demonstrates the importance of these nodes (i.e., the prediction of how many samples were affected by these nodes).

### 2.6. Software

We performed all analyses in publicly available custom-written Python-scripts. We used the following standard software packages for different functionalities within the custom scripts: Python v3.9.13 [42], pandas v1.4.4 [43], numpy v1.23.3 [44], sci-py v1.9.1, sklearn 1.1.2 [45], and matplotlib v3.5.2 [46]. Statistical testing was done using sklearn and penguin.

The presented algorithm is available as an open-source, ‘out-of-the-box’ functioning model including a detailed instruction [35]. We included a summary of the algorithm’s user instructions in the Appendix A.

### 2.7. Code and Data Availability

ReTap’s full algorithm is publicly available under MIT-license at www.github.com/jgvhabets/ReTap (accessed on 27 April 2023) [35]. Analysis scripts are available under MIT-license at www.github.com/jgvhabets/updrsTapping_repo (accessed on 27 April 2023).

Pseudonymised accelerometer data and labels will be made available after reasonable request to the corresponding author.

## 3. Results

### 3.1. Study Population and Recorded Data

We included 38 PwP in total, 20 in Düsseldorf and 18 in Berlin. Due to variations in the acquisition protocols across sites (i.e., the number of tapping blocks performed per subject), 66 tapping observations from the Düsseldorf subjects and 313 from the Berlin subjects, resulting in a total of 379 10-s tapping blocks, were included for further analysis. However, 29 accelerometer traces were excluded from the current analysis due to technical recording issues or incomplete data. This resulted in the inclusion of 350 tapping blocks in the final predictive analysis, with a tapping score distribution of 0:11.6%, 1:42.2%, 2:30.9%, and 3:15.3%.

The balanced data split led to 248 included traces originating from 26 subjects in the development dataset and 102 included traces originating from 10 subjects in the validation dataset. Stratifying for tapping score and center of origin caused a small deflection of the 75%/25% data split. Each data split contained equivalent distributions of expert-rated UPDRS scores compared to the total data set, and importantly, did not differ from one another (development data: 0:11%, 1:43%, 2:30%, and 3:16%; validation data: 0:12%, 1:42%, 2:33%, and 3:14%; F = 0.25, *p* = 0.617).

### 3.2. Automated Tapping Block and Single Tap Detection

The automated tapping block detection algorithm successfully detected 10-s tapping blocks with a sensitivity of 99.5% (i.e., 377 detected tapping blocks out of 379; Figure 1), which corresponded well with the onset and offset of tapping sequences based on visual inspection. However, we excluded 21 automatically detected tapping blocks as false positives based on visual inspection, leading to a positive predictive value of 94.7% (see Figure 1). Detected tapping blocks had a mean duration of 11.8 s (standard deviation (sd): 2.5 s). On average, a tapping block consisted of 29.5 (sd: 13) detected taps.

### 3.3. Finger Tapping Score Prediction

The holdout validation analysis of predicting expert-rated UPDRS Part III Item 3.4 scores from accelerometer-based kinematic features showed relatively good predictive performance, significantly better than chance level (i.e., 25%, see Methods 2.5), with a mean tapping score error of 0.56 (sd: 0.65, *p* < 0.001) and an ICC of 0.62 (*p* < 0.001) (see Figure 2, left panel) [47]. The true and predicted scores correlated moderately (Pearson’s r = 0.46, *p* < 0.001). The final selected model obtained features over the first 15 taps detected. For full summaries of model performance for the holdout dataset based on partial or total numbers of detected taps (i.e., results based on first 15 taps vs. all taps), see Appendix A.

On an individual subject level, the holdout validation showed a positive correlation between predicted and true scores in five out of seven subjects (71%) with calculated correlations (see Figure 2 and Appendix A). Of note, we could not calculate correlation coefficients for three subjects in which equivalent true UPDRS Part III Item 3.4 scores were observed regardless of recording/therapeutic session. Interestingly, two subjects with small numbers of included tapping blocks (i.e., three and four observations) exhibited moderate negative correlations between true and predicted UPDRS Part III Item 3.4 scores (see Figure 2 and Appendix A).

The kinematic features with the greatest importance for the RF classifier were full block jerkiness, the impact-RMS coefVar, the mean raise velocity, the full block normalized RMS, and the ITI coefVar as evidenced by larger mean decrease impurity scores per metric (see Appendix A). Since the individual analysis of the holdout results only considers a subset of the total included cohort, we additionally analyzed the sensitivity of ReTap’s kinematic features on the total cohort. We included the five most important features (mentioned above) and assessed their mean differences between individual therapeutic conditions (see Appendix A). We showed significant differences between individual best ON-conditions and medication-OFF or stimulation-OFF conditions for three out of five features (normalized RMS of full trace, jerkiness of full trace, and the mean finger-open velocity, *p* < 0.001). The normalized RMS values increased, the mean finger-opening velocity increased, and the coefficients of variation of inter-tap-intervals decreased under better therapeutic conditions (e.g., ON medication or stimulation) as expected. Additionally, while trace-jerkiness and the coefficient of variation of the RMS values were expected to be higher in worse therapeutic conditions (e.g., OFF medication or stimulation), these features demonstrated higher values under better therapeutic conditions. The latter might be explained by too high sensitivity of these features for the overall quantity of movement.

### 3.4. Feature Extraction

As an additional output, ReTap provides all extracted kinematic features per detected tapping block from preprocessed accelerometer traces. Figure 3 shows the course of four single-tap features in two exemplary subjects. It shows the differences in feature-course between tapping blocks with different expert-rated, true scores. For this example, we display data for four single-tap kinematic features that were relevant for the RF classification (impact-RMS, finger raising-velocity, ITI, and entropy per tap, see Appendix A). Expected differences in impact-RMS and raising-velocity are visible between the tapping scores, and in some of the cases, we observe a decrement over time characteristic for PD. Both the tap-entropy and the ITI were higher and more variable in tapping blocks, with worse tapping performance as expected.

## 4. Discussion

In the current study, we describe the development and validation of ReTap, a fully automated, open-source algorithm to predict index finger-to-thumb tapping scores (UPDRS Part III Item 3.4) based on accelerometer recordings from the index finger. Importantly, ReTap successfully predicted finger tapping symptom severity significantly better than chance level in an unseen holdout validation dataset. Moreover, ReTap-predicted UPDRS scores were moderately associated with expert-rated UPDRS scores, with intra-individual fluctuations in motor symptoms observed based on ReTap-predicted outcomes in 71% of the validation cohort. Below we discuss the implications of these findings and the methods used herein for future applications of automatically assessing PD-specific symptom severity.

### 4.1. Predictive Performance of ReTap

Quantifying bradykinesia symptoms and their severity based on task-relevant and naturalistic wearable sensor approaches has been a topic of scientific and clinical interest for over 30 years [48,49]. However, despite the accessibility of low-cost accelerometers, more computational resources, and their potential value for clinical care, there is no validated open-source model currently available to assess PD-relevant motor fluctuations. This notion emphasizes the theoretical and practical challenges of implementing automated UPDRS scoring procedures, which may be partially attributable to the multidimensional nature of UPDRS finger tapping assessments (i.e., considering the amplitude, rhythmicity, and associated decrements), as well as the inherently subjective ratings, especially on single items [16,17,18,50].

ReTap’s classification performance of tapping-related bradykinesia is good compared to benchmark paradigms that predict single-item UPDRS Part III scores without using individual training data, as evidenced by a mean prediction error of 0.56 (scale ranging from 0 to 3) and an ICC between true and predicted UPDRS scores of 0.62. Other non-proprietary algorithms demonstrating better performance based their predictive modeling approaches on acc-data collected from more than one sensor [51], or used video-based motion caption methods [30,52,53,54]. The slightly better performance from video-based approaches (i.e., ICC = 0.79) may be explained, in part, by the lower noise levels expected in video-based data. In contrast, recent studies of non-proprietary keyboard-based finger tapping methods did not exceed ReTap’s predictive performance on the single task level [28,29,31].

Morinan et al. recently reported a promising step towards automated UPDRS assessment. Their commercially available video-based assessment predicted full body bradykinesia with good predictive performance for all symptom severities (i.e., ICC = 0.74) [32]. However, their single item predictive performance model was restricted to the classification of two binary classes (i.e., good and bad), making comparability with current gold standards in symptom indexing (i.e., along a 5-stage scale) more difficult. Thus, there remains a need to empirically and automatically categorize single-item UPDRS outcomes on the traditional 0–4 scale.

### 4.2. Clinical Relevance and Potential Future Implementation of ReTap

ReTap’s main objective to produce an automated, clinically relevant prediction of finger tapping performance has potential to be used as an out-of-hospital assessment to collect reliable, validated scores of bradykinesia symptom severity. Moreover, the predicted scores may improve upon current standards for PD symptom monitoring by providing objective assessments of finger tapping performance in clinical settings. However, it is important to note that our results and benchmark results from prior studies of automated finger tapping assessments do not suggest that these models will outperform in-person assessments by experienced raters or clinicians on the single-item level (Figure 2) [18,32,52]. Instead, we propose that ReTap may *complement* current gold standards of symptom indexing (e.g., UPDRS Part III Motor Examination), which may ultimately lead to more reliable, comprehensive outlooks of clinical impairment in the future.

To be successfully applied as an “out-of-hospital” motor assessment, the task, device, and algorithm all need to be valid, reliable, and easily accessible. The relatively good predictive performance of the classification model at both the group and individual subject level demonstrates ReTap’s potential significance for generating reliable symptom scores for single observations, but also for longitudinal fluctuations in motor function (e.g., changes in medication/stimulation state/disease progression). The expected positive correlation between predicted and true UPDRS Part III Item 3.4 scores in 71% of the subjects in our validation cohort and the significant individual feature differences between therapeutic conditions (see Appendix A), which suggests that ReTap may be sensitive enough to detect individual fluctuations in finger tapping performance. It should be noted that ReTap’s application for at home monitoring has to be tested in patients’ natural environment still, and that the data quality is expected to be less consistent when self-recorded in the natural environment. The collection of gold standard, parallel bradykinesia assessments will be a major challenge here, and self-reported outcomes may partly solve this challenge [55]. Due to the low costs of accelerometers and the potential easiness to self-record data, accelerometer-based bradykinesia assessments such as ReTap have the potential to capture intraday fluctuations (multiple assessments per day) over longer periods. This is an evident advantage compared to video-based assessments, which provide scores with higher accuracy, but can be repeated less frequently due to technical difficulties in the recording set-up [32,52].

In addition to generating important task-related information regarding bradykinesia symptom severity, finger tapping scores may also be relevant for the further development of naturalistic passive sensing algorithms. Passive bradykinesia monitoring in shorter windows (e.g., minutes to an hour) is challenging [20,21]. Moreover, the passive prediction of bradykinesia severity in naturalistic settings seems to be more challenging than tremor and dyskinesia prediction [19,20]. A recent large in- and out-of-hospital trial showed that the reliability of passive measurements decreased with smaller time windows of assessments [21]. Additionally, in a prior study we were able to differentiate medication-ON vs. –OFF conditions on a minute basis, but could not predict bradykinesia severity [20]. Therefore, studies with more data containing task-relevant, short-windowed assessments of bradykinesia (i.e., finger tapping) are needed to provide true labels that can be used as ‘ground truth’ reflections of symptoms and their severity in order to aid the development of passive bradykinesia algorithms. Active monitoring of finger tapping performance as demonstrated using the ReTap algorithm may have the potential to fill this gap and to help overcome the current limitations of passive monitoring.

Furthermore, ReTap has the potential to improve finger tapping assessments in and out of the clinic by providing objective kinematic features of tapping performance at the single-tap and task-averaged level. This was an important consideration when developing ReTap in order to enable investigators to conduct more comprehensive, specialized analyses of clinically relevant tapping features in PwP. Other recent studies underscored the relevance of finger tapping analyses to assess bradykinesia and general motor improvement related to therapeutic outcomes. For example, in a prior study by Spooner et al. [56], the authors could demonstrate performance-related differences in movement kinematics (e.g., impact RMS, impact RMS coefVar, ITI, ITI coefVar) based on the direction of current administration within the subthalamic nucleus in PwP implanted with STN-DBS. Similarly, Feldmann et al. used accelerometry trace RMS values to detect motor performance differences explained by increasing subthalamic DBS amplitudes [57]. Detailed finger tapping analyses based on tapping speed, frequency, and variability were used as well to assess novel pharmacological PD treatment by Page et al. [58].

### 4.3. Importance of the Model’s Fully Automated and Open-Source Nature

The translation of model development and validation to clinical impact is notoriously difficult. To optimize the clinical impact of ReTap, we ensured that no signal preprocessing is required, and that the algorithm is publicly available. This will increase the accessibility of ReTap’s methods and the reproducibility of our results in future study cohorts implementing this approach. Importantly, the overwhelming number of sensor types, algorithms, and kinematic features used to assess bradykinesia-related deficits currently threatens the reproducibility of sensor-based finger tapping models [27]. Additionally, our methods may potentially improve the reproducibility of hand activity monitoring in a broader scope than merely UPDRS Part III Item 3.4 finger tapping performance alone. It is likely that our automated tapping activity detection and single tap detection functions may be useful for similar hand movement tasks (e.g., pronation-supination movements, open-close flexion-extension palm movements) and for other datatypes capturing hand movement (e.g., video-based motion caption), albeit future investigation is required in this area. Lastly, the automated generation of full feature time series per tapping sequence makes ReTap a useful toolbox for clinicians and researchers in neurology and movement studies.

### 4.4. Limitations

Our study is subject to several limitations. First, our holdout dataset contained 102 10-s tapping blocks originating from only 10 different subjects. A total of three of these ten subjects did not have any variability in their tapping scores. Although the chosen study design (i.e., including a holdout validation stratified for subjects) maximizes the validity of our predictive analysis, this number is relatively low. However, the large total sample size still allows for validated conclusions about ReTap’s predictive performance. Future reproduction of our results in new cohorts, ideally from other centers, would be valuable, nevertheless.

Second, the unbalanced nature of UPDRS Part III Item 3.4 tapping scores ranging from 0 to 4 across our sample may hinder certain predictive performance assessments. However, this imbalance aligns well with the natural distribution of bradykinesia severity commonly observed in PD populations. We ensured the same distribution in the development and holdout data in order to maximize statistical validity of the model results. We selected the reported predictive metrics (see Section 2.5) due to their applicability for unbalanced datasets. The minimal amount of UPDRS 4 scores observed in our dataset (<0.3%) prevented a validated detection of 4′s in the current study. We included a pragmatic solution for future applications of ReTap to identify tapping blocks containing barely any movement/detectable taps as UPDRS scores of 4. Among the tapping blocks with less than nine taps, the open-source model selects the blocks with RMS values lower than the 10th percentile of the tapping blocks with too few taps in the cross-validation data and classifies them as 4 s. This approach is pragmatic, although it is not validated in the current study cohort, but instead uses the validated detection of 3 s, which reflects the UPDRS assessment instructions.

Lastly, our cohort consists of subjects recruited and recorded at two different movement disorder clinics. Although subjects performed finger tapping paradigms with similar instruction sets, minor differences in acquisition protocols were inevitable. To account for site-related differences in the current study, we equally stratified subjects based on the recording site in both development and holdout datasets. However, ReTap’s performance, despite site-related factors, argues for a site-independent predictive performance, which is required for a real-world clinical implementation.

## 5. Conclusions

ReTap is a fully automated open-source tool to assess 10-s UPDRS Part III Item 3.4 finger tapping tasks based on index finger accelerometer data. We described ReTap’s algorithm to detect tapping blocks and single taps based on accelerometer data from the index finger and to predict expert-rated tapping scores. We validated its predicted scores by showing good predictive performance in a holdout validation dataset.

ReTap can provide objective, in-hospital finger tapping scores, including kinematic features for an in-detail tapping analysis. Moreover, ReTap has the potential to collect unsupervised, longitudinal finger tapping scores in an out-of-hospital environment. The future out-of-hospital application requires at-home validation but holds potential to provide validated bradykinesia estimates multiple times per day that can inform clinicians about intraday motor fluctuations. The latter could also provide repetitive predicted tapping scores that can function as ground truth labels for the development of passive bradykinesia monitoring.

## Figures and Tables

**Figure 1 sensors-23-05238-f001:**
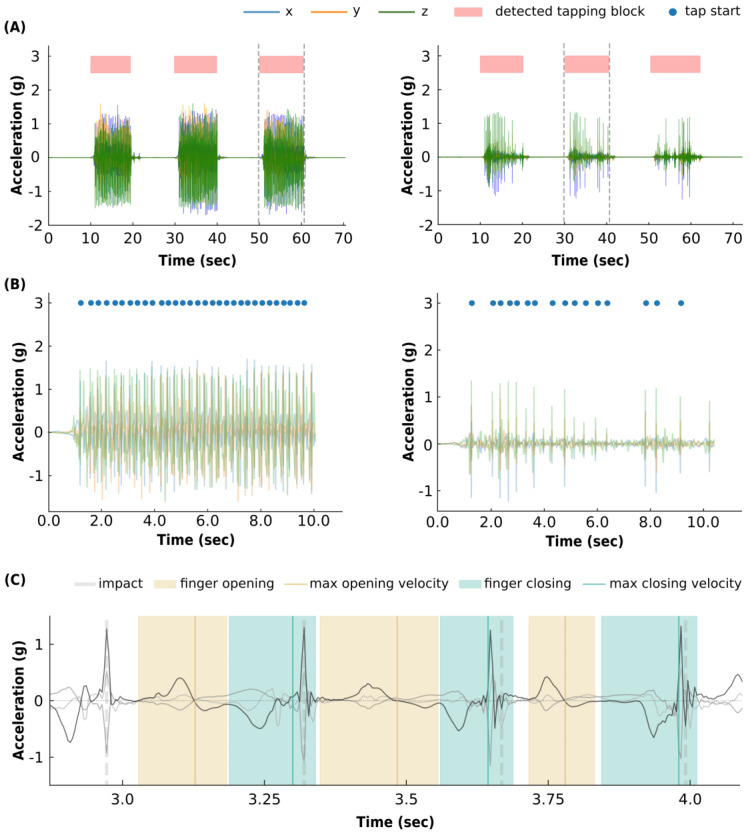
Automated finger tapping detection functions. (**A**): The automated tapping block detection results in two exemplary accelerometer traces containing three 10-s blocks of tapping activity. The function successfully detects repetitive 10-s tapping blocks present in the tri-axial accelerometer data, highlighted as the red blocks. The function performs well for taps with high (left panel) and low (right panel) amplitudes. (**B**): The automated single tap detection, performed on the tapping block between the dotted lines in the panel above. The blue dots represent the time points that the function detected impacts, which are used to recognize the moment of index finger and thumb closing. (**C**): Exemplary accelerometer trace snippet highlighting the temporal time points used for single tap feature extraction. Yellow shades indicate index finger opening and light-blue shades indicate index finger closing. The vertical yellow and blue lines indicate the moments of maximum speed within the finger opening and closing, respectively. Finger opening speed increases until the positive peak (in g) crosses 0 (vertical yellow line). Similarly, finger closing (downwards movement) speed increases during the negative acc-peak until the acc-signal crosses 0 g (vertical blue line). The vertical gray dotted lines represent the impact moment detected. The three accelerometer axes are shown in black and gray.

**Figure 2 sensors-23-05238-f002:**
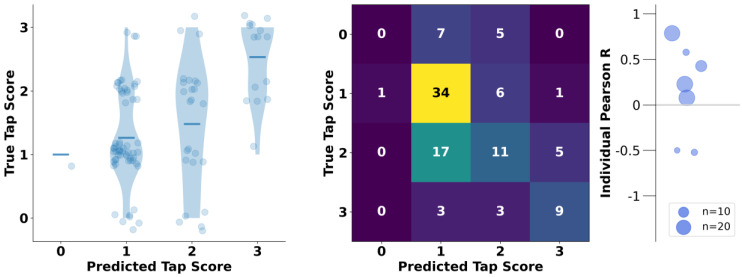
Prediction of finger tapping scores (UPDRS Part III Item 3.4) in the holdout validation. Left panel: Violin plots (with jittered scatter points representing one tapping block each) demonstrate single predicted tap scores versus true, expert-rated UPDRS Part III Item 3.4 scores. The horizontal lines represent the mean true UPDRS Part III Item 3.4 score per predicted tap score across the holdout validation sample. Middle panel: Multiclass confusion matrix showing prediction results per true UPDRS Part III Item 3.4 tap score during holdout validation. Right panel: Individual Pearson’s coefficients between the expert-rated scores and the ReTap-predicted scores per individual subject within the holdout validation (i.e., for which a correlation coefficient could be calculated). The dot size represents the number of tapping observations included per subject. See Appendix A for the individual subject-level observations leading to these correlations.

**Figure 3 sensors-23-05238-f003:**
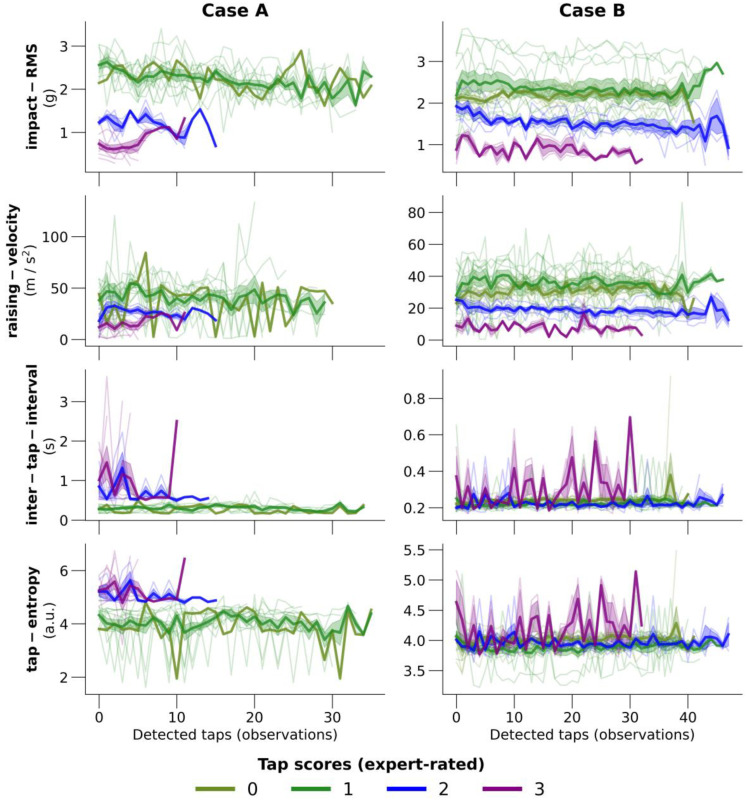
Exemplary cases of the kinematic features with the highest predictive performance. A total of two subjects from the holdout validation cohort are shown, each in one column. The four features are chosen based on the random forest feature importance (see Appendix A). Every thin line represents the feature values during one tapping block. Lines have various lengths of observations due to the various number of detected taps per tapping block. The thick lines represent the mean values of detected taps within tapping blocks of the same expert-rated score (i.e., mean value of first tap values in blocks with score 1, mean value of second tap values in blocks with score 1, etc.). Line colors indicate the expert-rated tapping score; olive green: 0, dark green: 1, blue: 2, purple: 3. ITI: inter-tap-interval.

## Data Availability

Pseudonymised accelerometer data and labels will be made available upon reasonable request.

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
