# Peer review of "A First Methodological Development and Validation of ReTap: An Open-Source UPDRS Finger Tapping Assessment Tool Based on Accelerometer-Data"

_sensors, 2023, doi:10.3390/s23115238_

Round 1

Reviewer 1 Report

Journal: Sensors (ISSN 1424-8220)

Manuscript ID: sensors-2398488

Type: Article

Title: Development and validation of ReTap: an open-source model for automated UPDRS finger-tapping assessment based on accelerometer-data

Summary:

The article is about the development and validation of an open-source algorithm called ReTap, which provides automated bradykinesia assessment in people with Parkinson's disease (PwP) using a finger-tapping task. The article explains the current challenges and limitations of existing methods for assessing motor symptoms in PD, and highlights the need for objective, reliable, and accessible tools for motor symptom monitoring. The article also discusses the potential benefits of technology-enabled motor monitoring, including the collection of other physiological data via sensing-enabled devices, and the reduction of lengthy in-person clinical visits and labor-intensive training periods currently required for passive monitoring.

Contribution to Existing Literature: 

The study highlights the potential of ReTap's repetitive predicted tapping scores as ground truth labels for passive bradykinesia monitoring. This aspect opens up avenues for the development of improved algorithms in monitoring bradykinesia over time. While the findings are promising, there are some important considerations to address. Specifically, the study would benefit from further elaboration on the algorithm's detection methodology and the validation process for out-of-hospital data collection. Additionally, the authors should acknowledge the potential limitations associated with relying on self-reported outcomes.

Moving forward, it is crucial to refine and enhance ReTap's algorithm based on the identified limitations and feedback from real-world applications. Ongoing studies and future validations in out-of-hospital settings will provide valuable insights into ReTap's effectiveness and address the remaining challenges.

Questions and comments on the Introduction Section:

1. The introduction states that Parkinson's disease (PD) is a neurodegenerative movement disorder expected to increase substantially in the coming years. Can you provide some specific statistics or references to support this claim?

2. Bradykinesia and akinesia are mentioned as cardinal symptoms of PD that negatively impact the quality of life in patients. It would be helpful to provide more context on the prevalence and severity of these symptoms in PD patients. Are there any studies or data that demonstrate the extent of their impact?

3. The introduction highlights that long-term dopamine replacement therapy leads to motor fluctuations in approximately 50% of PD patients. Could you explain in more detail how dopamine replacement therapy contributes to motor fluctuations? What are the implications and challenges associated with these fluctuations?

4. It is mentioned that automated, technology-based solutions for characterizing clinical features of bradykinesia would substantially improve the reliability, reproducibility, and accessibility of motor symptom assessments in PD. Can you elaborate on the current limitations of assessing bradykinesia using subjective clinical ratings? How would the development of automated tools address these limitations?

5. The introduction mentions the Unified Parkinson's Disease Rating Scale (UPDRS) as an example of a subjective clinical rating scale used for bradykinesia assessments. Could you provide more information on how the UPDRS is currently employed in clinical practice? What are the advantages and limitations of using the UPDRS for assessing bradykinesia?

6. The ideal motor assessment tool for PD is described as providing reliable symptom severity scores per category without requiring structured motor tasks. How does ReTap, the proposed algorithm, fulfill these criteria? What are the specific features or capabilities of ReTap that make it an effective tool for assessing bradykinesia?

7.  The introduction mentions the use of accelerometry, video-based motion capture, and keyboard- or smartphone-based tapping tasks as proposed methods for objective bradykinesia assessments. Can you provide more information on the strengths and limitations of each of these methods? How does ReTap compare to these existing approaches?

8. It is stated that there are currently no validated, automated, open-source accelerometer-based algorithms for bradykinesia assessment. Can you explain why accelerometer-based methods have not been validated or made available as open-source tools? What are the potential benefits of an open-source algorithm like ReTap in this context?

9. The introduction describes the specific goals and features of ReTap, including the detection of tapping activity, extraction of kinematic features, and prediction of UPDRS item 3.4 finger-tapping scores. Can you provide more information on why UPDRS item 3.4 finger-tapping assessments were chosen as the focus for ReTap? What evidence supports the idea that finger-tapping performance reflects general bradykinesia-related impairments?

10. Finally, the introduction mentions that ReTap has the potential to support accessible, out-of-hospital motor fluctuation monitoring for Parkinson's disease patients. Can you discuss the practical implications and potential benefits of implementing ReTap for remote monitoring? How could this impact the current clinical care and management of PD patients?

Questions and comments on the Materials and Method Section:

1. In the Study Sample section, it is mentioned that patients with PD from two academic movement disorders clinics in Düsseldorf and Berlin were included in the study. However, it would be helpful to know the total number of patients included in each clinic.

 2. In the Accelerometer data recording protocol section, it is stated that patients performed a unilateral finger-tapping task for ten seconds. Could you provide more information about the instructions given to the patients during the finger-tapping task? Were they instructed to tap at a specific rhythm or speed?

3. In the ReTap algorithm section, it is mentioned that the algorithm consists of five major parts. However, the description provided in the Materials and Methods does not include the details of all five parts. It would be helpful to provide a brief overview of the remaining parts of the algorithm.

4. In the Raw accelerometer data preprocessing section, it is mentioned that the raw tri-axial time series was resampled to 250 Hz if necessary. Could you clarify why resampling was necessary and how the decision to resample was made?

5. In the Active tapping block detection section, it is described how the algorithm detects active tapping windows. However, it would be helpful to know how the activity threshold (SVM standard deviation * 0.5) was determined and if it was based on any previous studies or empirical observations.

6. In the Single tap event detection section, it is mentioned that the algorithm detects tapping movements based on the closing of the index finger and thumb. Could you provide more information on how the algorithm identifies these moments in the accelerometer data? Are there any specific criteria used to determine the closing of the index finger and thumb?

7. In the Kinematic feature extraction per tapping block section, it is mentioned that several kinematic features are extracted. Could you provide more information about the rationale behind selecting these specific features? Were they chosen based on previous studies or established metrics for assessing motor symptoms?

8. In the Development and validation of the tapping score prediction model section, it is mentioned that the dataset was split into a development (75%) and validation (25%) dataset. Could you explain the rationale behind this split and how it ensures the statistical validity and reproducibility of the predicted UPDRS tapping scores?

9. In the Statistical Evaluation section, it is stated that the predictive performance of the classification model is reported in terms of mean prediction error, Intraclass Correlation Coefficient (ICC), and Pearson correlation coefficient. Could you provide the specific values or ranges for these metrics to assess the performance of the model more accurately?

10. It would be helpful if the article could provide more details about the statistical tests used for significance testing and the significance level chosen.

11. Overall, the Materials and Methods section provides a detailed description of the study design, data collection, and analysis procedures. However, providing additional information and clarification on the points mentioned above would enhance the understanding of the methodology.

Questions and Comments on the Result Section:

1. Can you explain the concept of tapping score and its distribution in the study population?

2. What was the data split for the development and validation datasets? How did it affect the distribution of expert-rated UPDRS scores?

3. What was the sensitivity of the automated tapping block detection algorithm?

4. Which kinematic features were found to be most important for the random forest (RF) classifier? Why?

5. How did the predicted and true scores correlate on an individual subject level?

6. What were the differences in kinematic features between tapping blocks with different expert-rated scores?

7. Can you explain the significance of the multiclass confusion matrix and the individual Pearson's coefficients in the holdout validation?

8. What additional output does ReTap provide in terms of feature extraction? How promising is it?

9. Can you clearly and scientifically explain the meaning and relevance of the features shown in Figure 3?

Questions and Comments on the Discussion Section:

1. In the discussion, the authors mention that ReTap's predictive performance is good compared to benchmark paradigms. Can you provide more details about these benchmark paradigms and how ReTap compares to them in terms of predictive accuracy?

 2. The authors state that ReTap may complement current gold standards of symptom indexing, such as the MDS-UPDRS. Could you elaborate on how ReTap can be integrated with existing assessment methods and how it can enhance the reliability and comprehensive outlook of clinical impairment?

 3. The potential implementation of ReTap as an "out-of-hospital" motor assessment is discussed. Have the authors considered the potential limitations or challenges of using ReTap in a natural environment? How would the collection of gold standard assessments be addressed in this context?

 4. The authors mention that ReTap has the potential to capture intraday fluctuations in motor function due to the low-cost and easy accessibility of accelerometers. Can you explain how ReTap's use of accelerometers differentiates it from video-based assessments and how it contributes to more frequent assessments?

 5. The discussion highlights the importance of ReTap in developing naturalistic passive sensing algorithms for bradykinesia monitoring. Could you provide more information on the challenges of passive prediction of bradykinesia severity and how ReTap can help overcome these limitations?

 6. The authors mention that ReTap provides objective kinematic features of tapping performance at the single-tap and task-averaged level, enabling specialized analyses. Could you provide examples of how researchers or clinicians can utilize these features and how they contribute to the assessment of bradykinesia?

 7. The authors emphasize the importance of ReTap's fully automated and open-source nature. Can you explain how the accessibility and reproducibility of ReTap's methods contribute to its clinical impact and improve hand activity monitoring beyond finger-tapping performance alone?

 8. The study acknowledges several limitations, including the relatively low number of subjects in the holdout dataset and the unbalanced nature of UPDRS tapping scores. Can you comment on how these limitations may impact the generalizability and validity of the study's findings?

 9. The authors mention that the cohort consists of subjects recorded at two different movement disorder clinics. Could you elaborate on how the site-related differences were accounted for in the study and how they may have influenced ReTap's performance?

Questions and comments on the Conclusion Section:

1. Are there any plans or ongoing studies to validate ReTap in a real-world out-of-hospital environment? If so, what are the specific objectives and methodologies of these studies?

2. Are there any plans to further improve or refine ReTap's algorithm based on the findings and limitations identified in this study? If so, what areas of improvement are being considered?

Author Response

Please see the attachment for point-by-point responses to the reviewer's comments highlighted in blue text. 

Reviewer 2 Report

The manuscript is well written and well structured. The text is clear and easy to read. The topic is interesting and in line with the journal.

Introduction

The introduction of the study is well structured, the rationale behind the study is written in a clear and understandable way, moreover it includes the main aims of the study

Materials and Methods

This section contains enough information to understand and possibly repeat the study. Almost every aspect of the study has been considered and explained in detail but inclusion and exclusion criteria were  not described and listed. Authors should write inclusion and exclusion criteria and the characteristics of the study sample (Hoehn and Yahr scale, age, male/female, etc..)

The small number of the sample analyzed does not support the conclusions. should be considered as a pilot study or a methodological study and accordingly change the title to “A first methodological study of development and validation of ReTap: an open-source model for automated UPDRS finger tapping assessment based on accelerometer-data” 

Author Response

(The authors gave the same response as above.)

Round 2

Reviewer 1 Report

It can be published in its present form.